Corrected: Author correction

# Ultra-thin enzymatic liquid membrane for $CO_2$ separation and capture

Yaqin Fu[1,2], Ying-Bing Jiang[1,2,3], Darren Dunphy[1,2], Haifeng Xiong [1,2], Eric Coker [4], Stanley S. Chou[4]
Hongxia Zhang[5], Juan M. Vanegas[4,6], Jonas G. Croissant[1,2], Joseph L. Cecchi[1], Susan B. Rempe [4] &
C. Jeffrey Brinker[1,2,4]

The limited flux and selectivities of current carbon dioxide membranes and the high costs associated with conventional absorption-based $CO_2$ sequestration call for alternative $CO_2$ separation approaches. Here we describe an enzymatically active, ultra-thin, biomimetic membrane enabling $CO_2$ capture and separation under ambient pressure and temperature conditions. The membrane comprises a ~18-nm-thick close-packed array of 8 nm diameter hydrophilic pores that stabilize water by capillary condensation and precisely accommodate the metalloenzyme carbonic anhydrase (CA). CA catalyzes the rapid interconversion of $CO_2$ and water into carbonic acid. By minimizing diffusional constraints, stabilizing and concentrating CA within the nanopore array to a concentration 10× greater than achievable in solution, our enzymatic liquid membrane separates $CO_2$ at room temperature and atmospheric pressure at a rate of 2600 GPU with $CO_2/N_2$ and $CO_2/H_2$ selectivities as high as 788 and 1500, respectively, the highest combined flux and selectivity yet reported for ambient condition operation.

[1] Department of Chemical and Biological Engineering, University of New Mexico, Albuquerque, NM 87131, USA. [2] Center for Micro-Engineered Materials, University of New Mexico, Albuquerque, NM 87131, USA. [3] Department of Earth and Planetary Sciences, University of New Mexico, Albuquerque, NM 87131, USA. [4] Sandia National Laboratories, Albuquerque, NM 87185, USA. [5] Angstrom Thin Film Technologies LLC, Albuquerque, NM 87113, USA. [6] Department of Physics, University of Vermont, Burlington, VT 05405, USA. Correspondence and requests for materials should be addressed to Y.-B.J. (email: ybjiang@unm.edu) or to C.J.B. (email: cjbrink@sandia.gov)

Carbon dioxide ($CO_2$) is the most important anthropogenic greenhouse gas in the atmosphere[1–3]. According to the 2014 report of the World Meteorological Organization[4], atmospheric $CO_2$ reached 142% of its pre-industrial level in 2013, primarily because of emissions from the combustion of fossil fuels and production of cement. In November 2016, the Paris Accord was ratified with the goal of maintaining a global temperature rise of only 2 °C above pre-industrial levels during this century. However, the realization of this goal is imperiled by the cost of $CO_2$ sequestration. Seventy percent of the cost of capturing of $CO_2$ involves separation from other gases.

The conventional process for $CO_2$ capture involves reversible absorption[3,5], which consumes high amounts of energy and is costly with a high environmental impact[3]. More efficient and environmentally friendly separation processes are needed, and in this context, membrane separation represents a promising approach due to its greater energy efficiency, processability, and lower maintenance costs[5–7]. Membranes enabling selective and efficient removal of $CO_2$ from fuel gas (containing CO, $H_2$, $H_2O$, and $H_2S$) or flue gas (containing $N_2$, $O_2$, $H_2O$, $SO_2$, NOx, and HCl) could be of great economic value[8]. An efficient membrane should have both high permeance and selectivity. Permeance is the flux of a specific gas through the membrane, typically reported in Gas Permeation Units (GPUs) (1 GPU = $10^{-6}$ cm$^3$ (STP) cm$^{-2}$ s$^{-1}$ cm$^{-1}$ Hg$^{-1}$). Selectivity is the capacity to separate two or more gases, typically reported as a dimensionless ratio of flux. Porous membranes usually exhibit a high $CO_2$ flux, but due to pore size variability, they often display a poor selectivity. Notable exceptions are zeolite membranes whose sub-nanometer pore size is defined by the zeolite crystallographic lattice and is monodisperse. Recently Korelskiy et al. reported an H-ZSM-5 zeolite membrane exhibiting a $CO_2/H_2$ selectivity of ca. 200 and a $CO_2$ permeance of ca. 17,000 when operated at 9 bars and −43 °C[9,10]. Dense membranes, typically polymers, exhibit moderate selectivity, but the $CO_2$ flux is usually low because of the low solubility and diffusivity of $CO_2$. In general, most existing membranes exhibit a sharp trade-off between flux and selectivity and are so far impractical for $CO_2$ capture applications[2,7,11,12].

Three factors govern membrane flux and selectivity: (1) how fast the species to be separated can enter into or exit from the membrane, (2) how selectively it can enter into or exit from the membrane, and (3) how fast it can be transported through the thickness of the membrane. Not surprisingly, biological systems maximize the combination of these factors as separation processes typically take place in an ultra-thin liquid layer aided by enzymes that catalyze the selective and rapid dissolution and regeneration of the target species (increasing solubility and selectivity), and short diffusion distances combined with higher diffusivity within liquid vs. solid media maximize transport rates. For $CO_2$ in particular, the respiratory system of vertebrates is an excellent case in point. Red blood cells employ carbonic anhydrase (CA) enzymes to rapidly and selectively dissolve the $CO_2$ produced by tissues and regenerate the $CO_2$ exhaled from the lung. CAs represent a family of metalloenzymes that catalyze the rapid interconversion of $CO_2$ and water into carbonic acid $H_2CO_3$ (Eq. 1), which dissociates to bicarbonate ($HCO_3^-$) and protons according to the prevailing species concentrations (Fig. 1 and Supplementary Fig. 1). Carbonic anhydrases are necessarily one of the fastest enzymes with reported catalytic rates ranging from $10^4$ to $10^6$ reactions per second, meaning that one molecule of CA can catalyze the hydration/dissolution of 10,000 to 1,000,000 molecules of $CO_2$ per second[13,14].

$$CO_2 + H_2O \Leftrightarrow H_2CO_3 \Leftrightarrow HCO_3^- + H^+ \qquad (1)$$

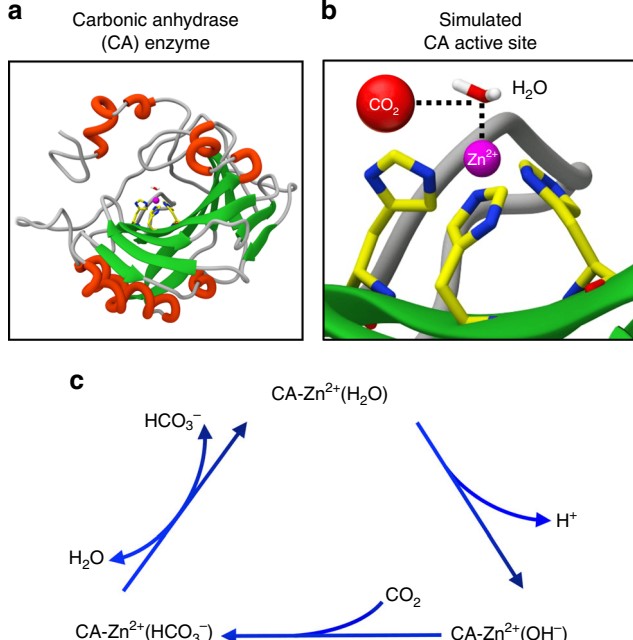

**Fig. 1** Carbonic anhydrase enzyme and its $CO_2$ capture and regeneration mechanism. **a** Ribbon representation of the carbonic anhydrase (CA) enzyme. **b** Active site of CA determined by molecular simulations (vide infra). A zinc ion ($Zn^{2+}$) surrounded by three coordinating histidines and a water molecule comprises the active site. **c** Depiction of the overall catalytic cycle for $CO_2$ hydration to $HCO_3^-$ with zinc as the metal in the CA active site. This reaction is driven by a concentration gradient: clockwise when the $CO_2$ concentration is greater than $HCO_3^-$ and counterclockwise when more $HCO_3^-$ is present. Deprotonation of the zinc-bound water is thought to be rate limiting

The concept of employing CA for $CO_2$ separation was first reported by Ward and Robb who impregnated a cellulose acetate film with a potassium bicarbonate solution containing CA and observed a factor of six increase in $CO_2$ permeability over potassium bicarbonate alone[15]. Based on a similar concept, Carbozyme Inc. encapsulated an aqueous CA solution within a microporous polypropylene hollow fiber membrane[15] and achieved a five times higher $CO_2$ permeability compared to Ward and Robb's membrane. However, the $CO_2$ flux (18.9 GPU[16]) still fell far short of that needed for practical $CO_2$ sequestration, since a $CO_2$ capture cost below $20–40 per ton is required by the U.S. Department of Energy[17], which translates into a $CO_2/N_2$ selectivity higher than 30–50 as well as a $CO_2$ permeance higher than 300–3000 GPU[17,18]. Inherent problems/limitations of CA membranes developed to date are thickness (10–100 μm,), which establishes the diffusion length and limits flux, and CA concentration, which governs the $CO_2$ dissolution and regeneration rates, but is limited in by the enzyme solubility (typically <1 mM).

Here, in order to overcome the limitations of current $CO_2$ membranes and exceed the DOE requirements for $CO_2$ sequestration, we have developed an ultra-thin, CA-catalyzed, liquid membrane nano-stabilized via capillary forces for $CO_2$ separation (see Fig. 2). It comprises oriented, close-packed arrays of 8 nm diameter hydrophilic cylindrical nanopores (silica mesopores[19]), whose effective thickness (i.e., the hydrophilic pore length/depth) is defined by oxygen plasma treatment to be ~18 nm. Through capillary condensation, the pores are filled with water plus CA enzymes confined and stabilized to high pressures by nanoconfinement (approximately the capillary pressure, ~35 atmospheres) exerted by water within a hydrophilic 18 nm diameter nanopore).

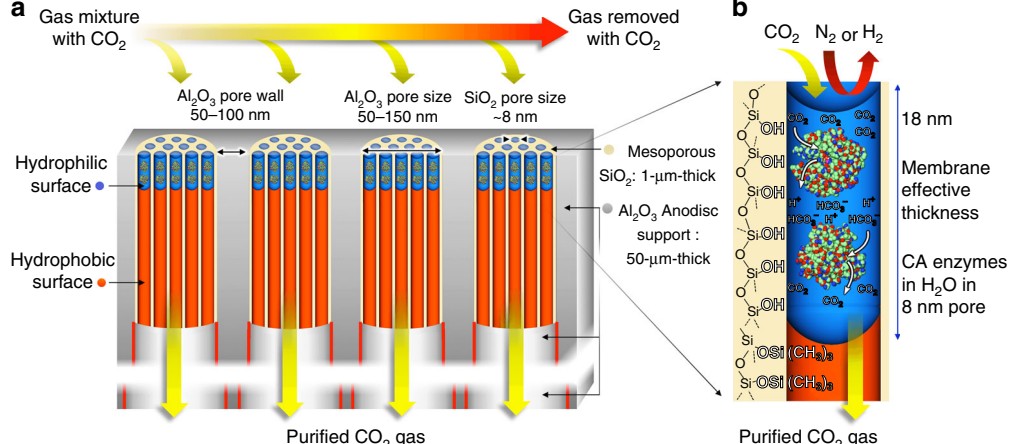

**Fig. 2** Enzymatic liquid membrane design and mechanism of $CO_2$ capture and separation. **a** The membrane is fabricated by formation of ~1-μm-deep oriented arrays of 8 nm diameter cylindrical silica [$SiO_2$] mesopores within the larger 50–150-nm pore channels of a 50-μm-thick porous alumina [$Al_2O_3$] Whatman© Anodisc support. **b** Using atomic layer deposition and oxygen plasma processing (described in text and Fig. 4) the silica mesopores are engineered to be hydrophobic (trimethylsilyl ($Si(CH_3)_3$) surface groups) except for an 18-nm-deep region at the pore surface, which is hydrophilic ($\equiv Si-OH$ surface groups). Via capillary condensation, CA enzymes and water spontaneously fill the hydrophilic mesopores to form an array of nano-stabilized CA enzymes with an effective CA concentration >10× of that achievable in solution. CA catalyzes the capture and dissolution of $CO_2$ as carbonic acid ($HCO_3^-$) moieties at the upstream surface and regeneration of $CO_2$ at the downstream surface (see Fig. 1c). The high concentration of CA and short diffusion path length maximizes capture efficiency and flux

Due to the exceptional thinness of the membrane and the high effective concentration of CA within the close-packed arrangement of nanopores, we demonstrate (under approximately ambient conditions of pressure and temperature) unprecedented values of combined $CO_2$ flux (as high as 2600 GPU) and $CO_2/N_2$ selectivity (as high as 788). Because the $CO_2$ selectivity derives from that of the confined CA enzyme, the enzymatic liquid membrane also exhibits high $CO_2/H_2$ selectivity (as high as 1500).

## Results

**Ultra-thin hydrophilic nanoporous membrane fabrication.** The enzymatic liquid membrane was fabricated using a four step-process (Figs. 3 and 4). Step 1 involved the fabrication of an architecture that both stabilizes water and can accommodate CA enzymes (vide infra). The oriented Anodisc pores were thus sub-divided into smaller, oriented, 8 nm diameter cylindrical pores via deposition of P123 block copolymer templated mesoporous silica using the so-called evaporation-induced self-assembly (EISA[20,21], see Methods). In this process, the Anodisc pore channels are filled to a depth of about 1 μm with a cylindrical hexagonal P123/silica mesophase (space group *p6mm*), which when confined to a cylindrical channel orients parallel to the channel axis (see Fig. 3c–f). Calcination at 400 °C is used to remove the P123 template resulting in oriented 8 nm diameter nanopores (see Fig. 3c, d) whose pore surfaces are terminated with hydrophilic surface silanol groups ($\equiv Si-OH$). Note that surfactant removal can be accomplished at room temperature by UV/ozone or oxygen plasma treatment[22]. Hydrophilic 8 nm diameter nanopores are large enough to accommodate CA (~5.5 nm in diameter) within a confined water layer and small enough to spontaneously fill with water above ~75% relative humidity (RH) (vide infra). However, the thickness of the resulting nano-stabilized liquid membrane would be ~1 μm far exceeding that of natural membranes. In order to reduce the effective thickness of the nano-stabilized liquid membrane, we conducted two steps of surface modification (Steps 2 and 3, Fig. 4). In step 2, using an atomic layer deposition (ALD) apparatus, we treated the membrane with ozone to maximize the surface silanol coverage and then conducted five cycles of alternating (hexamethyldisilizane

(HMDS) and trimethylchlorosilane (TMCS)) and $H_2O$ vapor exposures to quantitatively replace hydrophilic surface silanol groups with hydrophobic trimethylsilyl groups ($Si(CH_3)_3$). In step 3, we then exposed the membrane to a remote oxygen plasma for 5 s to re-convert hydrophobic trimethylsilyl groups to hydrophilic silanol groups at the immediate membrane surface. The mechanism of this plasma-nanopore-modification has been described by us previously[21,23]. Briefly, reactive radicals generated in a low-pressure oxygen plasma are mainly charged ions that cannot penetrate deeply into the nanoporous support, because the plasma Debye length (~20 cm under our conditions) is much larger than the pore size (~8 nm). In order to confirm the hydrophilicity of the plasma-modified nanoporous membrane surface and the hydrophobicity of the HMDS-modified surface, the water contact angle was measured with a Biolin Scientific Theta Optical Tensiometer. Fig. 5a shows the hydrophilic surface to have a contact angle of nearly 0° (note since the water droplet for the measurement is about 0.05 ml, not all water can be adsorbed in the nanopores, and some excessive free water remains on the surface) consistent with a superhydrophilic surface stemming from the hydrophilic surface chemistry and nanoscale roughness[24]. In comparison, the water contact angle of the HMDS-modified surface was ~150° consistent with a super-hydrophobic surface stemming from the hydrophobic surface chemistry plus nanoscale roughness[25].

In order to estimate the depth of the hydrophilic plasma-modified surface layer, we compared $TiO_2$ ALD on the original hydrophilic mesoporous silica membrane with $TiO_2$ ALD on the HMDS plus oxygen plasma-modified 'amphiphilic' membrane, using conventional $TiCl_4$ and $H_2O$ vapor as the $TiO_2$ ALD precursors. It is well established that $TiO_2$ ALD requires a hydrophilic (normally hydroxylated) surface to initiate deposition; therefore, the formation of $TiO_2$ can be used to 'map' the hydrophilic surface chemistry. Fig. 5b shows the EDS-based Ti elemental mapping of cross-sectional samples, where the brightness corresponds to the Ti concentration. The bottom row is a cross-section of the original mesoporous silica membrane, where we observe Ti deposition throughout the ~250-nm-thick section (membrane top surface is on top) as expected from the

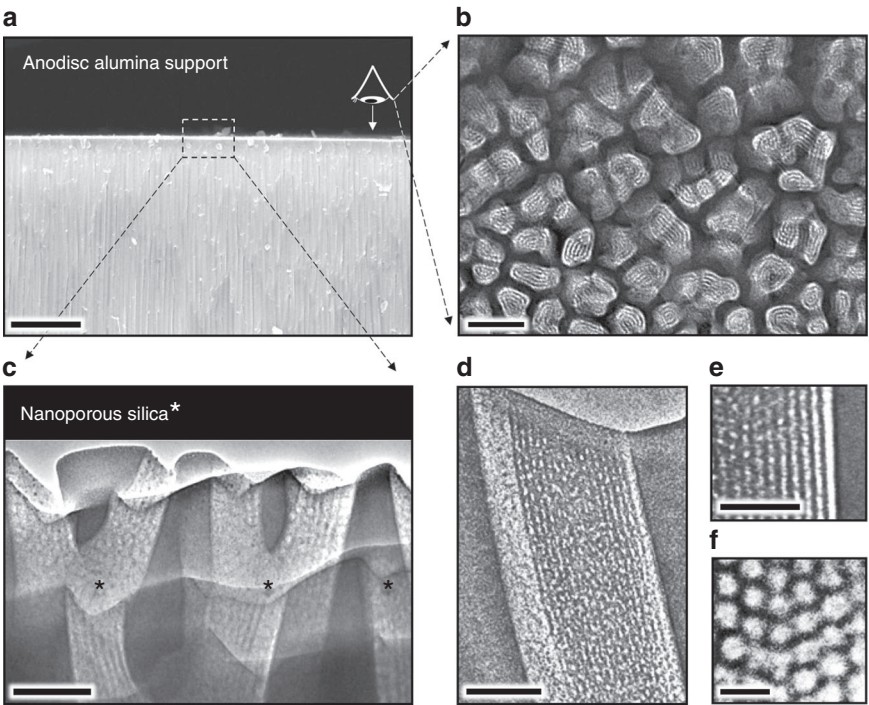

**Fig. 3** Electron microscopy images of the membrane hierarchical macro-structure and nano-structure. **a** Cross-sectional SEM image of the Anodisc support showing oriented ~50-nm-wide pore channels near the top surface (scale bar: 5 µm). **b** Plan-view TEM image of focused ion beam (FIB)-sectioned Anodisc surface showing complete filling of all Anodisc pore channels with ordered arrays of silica mesopores (scale bar: 100 nm). (Note: FIB sectioning served to etch the alumina leaving only the silica mesopore arrays. Silica mesopore arrays not perfectly aligned normal to imaging axis appear as stripe patterns). **c** Cross-sectional TEM image of the Anodisc surface showing oriented arrays of 8 nm diameter cylindrical mesopores filling the Anodisc pores (scale bar: 100 nm). **d**, **e** Higher magnification cross-sectional TEM image showing oriented array of 8 nm diameter cylindrical mesopores filling a single Anodisc pore (scale bar **d**: 100 nm; scale bar **e**: 50 nm). **f** Plan-view TEM image of silica mesopore array at membrane surface showing hexagonal close packing of cylindrical mesopores (scale bar: 20 nm)

hydroxylated surface chemistry. The top row shows that Ti ALD on the HMDS-plasma-modified amphiphilic membrane is confined to an ~18-nm-deep hydroxylated region on the immediate surface—this depth establishes the effective thickness of the confined liquid membrane to be only 18 nm (vide infra).

**Sub-20-nm-thick enzymatic liquid membrane fabrication.** Having successfully fabricated an ultra-thin hydrophilic nanoporous layer on the hydrophobic support, we next introduced CA enzymes into the hydrophilic nanopores by simple immersion of the sample in an aqueous enzyme solution with a CA concentration of 0.05 mM (Step 4, Fig. 4) After moderate bath sonication for 10 min, the sample was withdrawn from the solution and allowed to 'dry' in a horizontal configuration. During this evaporation process, the CA enzyme solution is concentrated and stabilized within the hydrophilic nanopores via capillary forces to form an ultra-thin liquid membrane containing CA enzymes. Since the superhydrophobic pores repel water, the thickness of the CA containing liquid membrane is defined by the thickness of the hydrophilic nanoporous layer, which was determined to be about 18 nm (Fig. 5b).

Direct observation of the formation and thickness of the liquid membrane is challenging. However, by measurement of the mass of water adsorbed within a defined area of the amphiphilic nanoporous membrane, we can calculate the effective liquid membrane thickness according to its geometry. In order to perform this experiment, we used a quartz crystal microbalance (QCM) to measure the mass of water adsorbed within the nanoporous membrane deposited onto the active area of the

QCM and processed identically to the membrane deposited on the Anodisc support, i.e., by HMDS/TMCS ALD followed by plasma processing. To confirm the structural similarity of the films deposited on the QCM and AO support surfaces, we performed grazing-incidence small-angle scattering (GISAXS). Fig. 6a, b compares the respective GISAXS data where we observe nearly identical patterns confirming the structural similarity of the samples. Then we introduced coated-QCM devices into an environmental chamber and performed water adsorption isotherms. Fig. 6d compares the $H_2O$ adsorption isotherms of nanoporous silica films processed before and after plasma processing, where 0% RH corresponds to samples purged using pure dry $N_2$ for more than 1 h. For the original HMDS/TMCS-treated hydrophobic nanoporous silica membrane (referred to as 'hydrophobic' in Fig. 6d), the mass of the sample shows a small increase with increasing RH, probably due to water vapor adsorption by randomly scattered hydrophilic micropores that are inaccessible to HMDS/TMCS molecules during ALD. For the membrane prepared by HMDS/TMCS ALD followed by plasma irradiation (referred to as 'amphiphilic' in Fig. 6d), the mass of water adsorbed increases abruptly at about 75% RH consistent with spontaneous water absorption by capillary condensation and the formation of the nano-stabilized liquid membrane (vide infra). The 4.82 µg mass increment at RH = 75% corresponds to a volume of $4.82 \times 10^{-6}$ $cm^3$ of water. Assuming a 50% volumetric porosity of the nanoporous silica membrane (as is typical for P123-templated mesoporous silica) and using the geometric surface area of 4.91 $cm^2$ for the 25 mm diameter QCM sensor, we calculate the corresponding water layer thickness to be 19.6 nm, which is in reasonable agreement with the 18 nm thickness

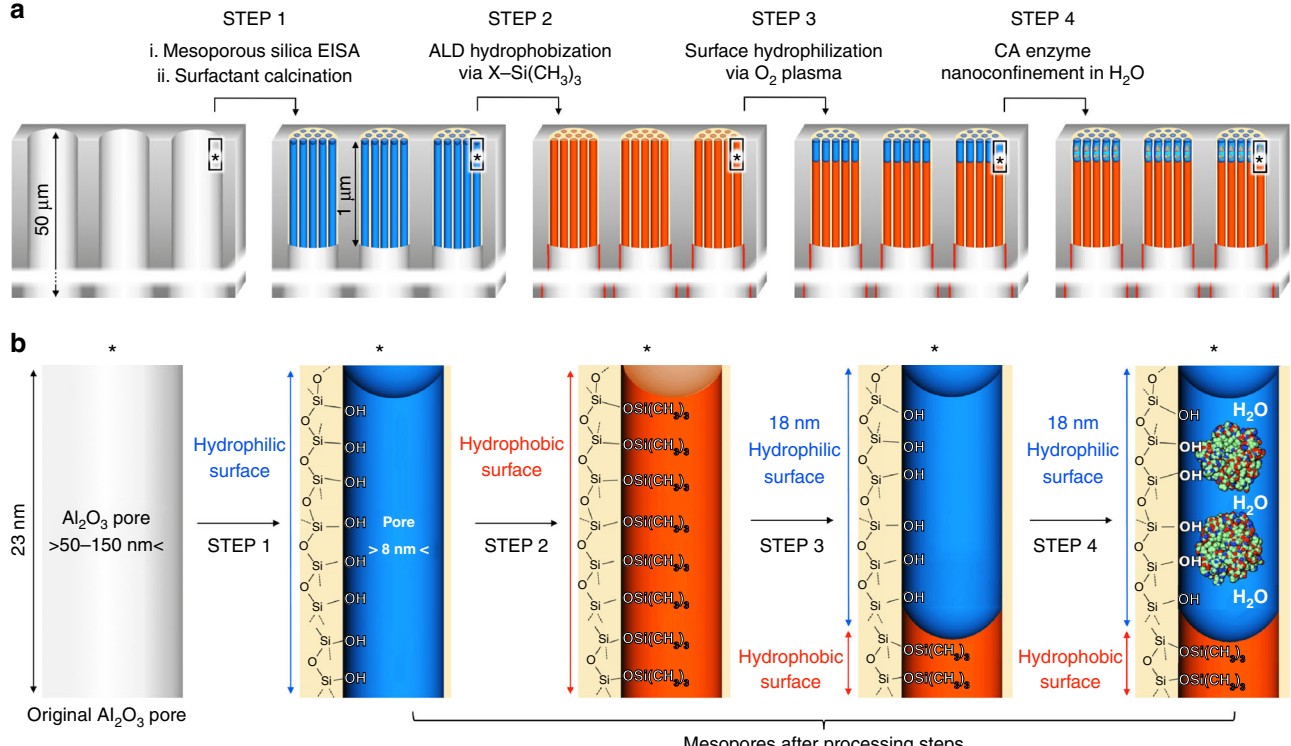

**Fig. 4** Design steps of the enzymatic liquid membrane. Beginning with a 50-μm-thick Anodisc support, Step 1 comprises the formation of oriented arrays of 8 nm diameter cylindrical silica mesopores within the 50–150 nm diameter Anodisc pores via evaporation-induced self-assembly followed by calcination to remove the P123 surfactant. In Step 2, three alternating cycles of atomic layer deposition (ALD) of HMDS ((CH₃)₃-Si-N-Si(CH₃)₃) + TMCS (Cl-Si(CH₃)₃) followed by water are conducted to convert the hydrophilic silanol-terminated mesopore surfaces to hydrophobic Si-O-Si(CH₃)₃ surfaces throughout the 1 μm length of the mesopore. In Step 3 a remote oxygen plasma treatment is used to regenerate hydrophilic silanol groups to a depth of 18 nm on the top surface. In Step 4 an aqueous solution of CA is introduced on the top surface. Through capillary condensation, water plus enzymes fill the mesoporous silica array. **a** images represent the processing steps and **b** images represent the corresponding surface chemistries

observed according to the TiO₂-ALD control experiments (Fig. 5b).

In order to prove the formation and the air-tightness of the liquid membrane, the permeance of N₂ (maintained at 95% RH) through the membrane (prepared as described above) was measured using a bubble flow rate meter for a 1 atm pressure difference. The permeance of N₂ through the membrane was almost undetectable, whereas, in contrast, the N₂ permeance through the completely hydrophobic sample (i.e., prepared without plasma irradiation, and thereby, having no stabilized water layer) was measured to be 340 sc cm cm⁻² atm⁻¹. As a further control, we also measured the permeance of CO₂ (maintained at 95% RH) through the membrane prepared as described above, but without the CA enzymes, i.e., through the ultra-thin stabilized water layer. In this case the CO₂ permeance was undetectable. These results indicate that the ultra-thin CA containing liquid membrane is continuous and essentially defect-free. One conceivable concern might be how to ensure that the liquid membrane is stable and will not 'dry out' in real-world applications. As previously discussed, this concern is alleviated by maintaining the membrane at a sufficient relative humidity where, due to capillary condensation, the uniformly sized hydrophilic nanopores remain water-filled. According to the Kelvin equation, capillary condensation for a hydrophilic pore occurs at a relative humidity RH defined by: $\ln(\text{RH}) = -(2\gamma V_m/rRT)$, where $\gamma$ and $V_m$ are the surface tension and the molar volume of water, $r$ is the pore radius, $T$ is the temperature in Kelvin, and the $R$ constant (8.32 J mol⁻¹ K⁻¹). For the 8 nm diameter pores of our membrane, the Kelvin equation predicts condensation to occur at an RH equal to or exceeding 75%, which

is consistent with the water adsorption 'step' observed in Fig. 6d. A typical flue gas comprises 6.2 wt% H₂O if it is from a coal-fired plant and 14.6 wt% H₂O if from a gas-fired plant. Both are much higher than the saturated water vapor concentration at 40 °C (~50 g H₂O kg⁻¹ air or 0.5 wt% H₂O). Therefore, the humidity requirement to maintain membrane stability can be easily satisfied if the membrane is used to capture CO₂ from power plant flue gas or used in any moderate humidity environment (see Supplementary Discussion).

Another potential concern is that of the liquid membrane strength, e.g., will the liquid membrane be ruptured when applying pressurized gas for separation? Here, the uniform nano-sized dimensions of the hydrophilic pores assure mechanical stability: the capillary pressure of water condensed within a pore can be calculated according $P = 2\gamma\cos\theta/d$ (where $\gamma$ is the water-air surface tension and $d$ is the pore diameter). For water confined within 8 nm diameter hydrophilic pores, where the contact angle $\theta$ equals zero, the capillary pressure is about 35 atm (Supplementary Discussion). Therefore, under regular operations like CO₂ capture from flue gas, where the gas pressure is typically less than several atmospheres, the capillary pressure is more than sufficient to stabilize the membrane and prevent its displacement into the hydrophobic portion of the membrane nanopores.

**Enzymatic liquid membrane performance.** So far, we have demonstrated an 'air-tight', ultra-thin, stable, enzyme-containing liquid membrane formed on an Anodisc support. Next, we measured the CO₂ permeance of the enzymatic liquid membrane fabricated with mammalian or extremophile CA enzymes at

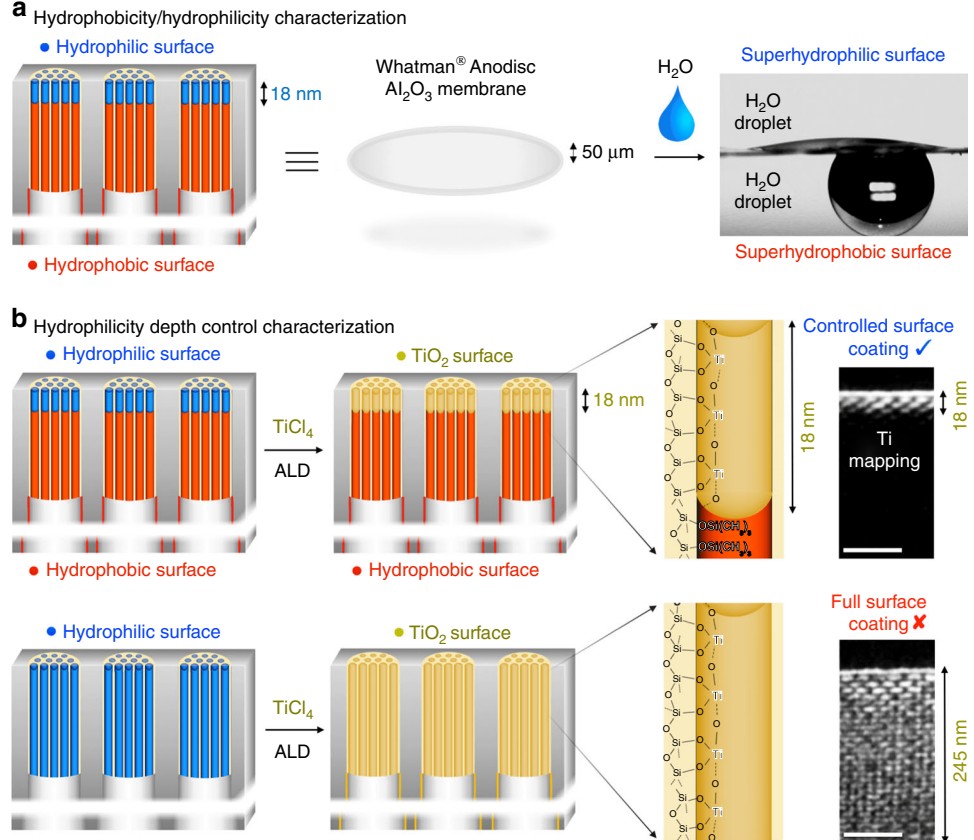

**Fig. 5** Hydrophilicity depth characterization of the enzymatic liquid membrane. **a** Representation of the opposing hydrophilic/phobic character of the membrane surfaces and corresponding water contact angle characterization. **b** Control experiments designed to probe the hydrophilicity depth of the 'amphiphilic' membrane. The hydrophilic surface modified membrane (top row) showed a limited (18-nm-deep) atomic layer deposition (ALD) of titanium oxide ($TiO_2$), mapping the depth of silanol groups (scale bar: 50 nm). In contrast, the fully hydrophilic membrane (bottom row) shows $TiO_2$ deposition throughout its thickness. The effective membrane thickness of the enzymatic water-membrane for $CO_2$ capture and separation is thus determined to be ~18 nm

various temperatures and pH values (Fig. 7a, b). We then determined and compared our experimentally observed $CO_2/N_2$ separation efficiency and $CO_2$ flux performance with other as-reported $CO_2$ membranes, and the corresponding data are plotted in Fig. 7f.

Figure 7a compares the $CO_2$ permeances at different temperatures for two types of CA enzymes: CA derived from mammalian bovine erythrocytes and CA derived from *Desulfovibrio vulgaris*—an extremophile bacteria that survives under conditions of 5 °C and pH 10. For the Bovine CA enzyme, the permeance, resulting from CA mediated $CO_2$ dissolution (Eq. 1) followed by diffusion across the 18-nm-thick liquid membrane and ex-solution (reverse of Eq. 1) at the hydrophobic interface is temperature dependent and, as expected, is maximized at mammalian body temperature, 30–40 °C. For membranes containing the *Desulfovibrio vulgaris* CA enzyme, the $CO_2$ permeance is practically temperature independent exceeding that of the bovine CA membrane at low and high temperatures, but found to be less than that of bovine CA at 30–40 °C. Our observed temperature dependent CA activity is in good agreement with that reported by Hooks and Rehm[26]. Fig. 7b plots $CO_2$ permeance as a function of pH for bovine CA membranes and *Desulfovibrio vulgaris* CA membranes. Similar to the temperature dependence, the bovine CA membranes performed best at neutral pH, whereas, the *Desulfovibrio vulgaris* CA membranes exhibited a

very moderate pH dependence over the pH range 2–10 and exhibited higher $CO_2$ permeance at both lower and higher pHs.

Figure 7f compares the $CO_2$ separation and permeance performance of our ultra-thin enzymatic liquid membrane to that of other classes of $CO_2$ membranes. The liquid membrane was operated at 37 °C and pH 7.5 with only a chemical potential driving force. The feed gas composition was 20 vol% $CO_2$ in $N_2$ maintained near ambient pressure (36 cm Hg (0.48 bar)) and the collection side comprised a $Ca(OH)_2$ aqueous solution to capture $CO_2$ and maintain a constant chemical potential driving force (see Methods and Supplementary Fig. 3 for setup). With the exception of the high permeance ZSM-5 membrane operated at 9 bars over the temperature range 37 to −43 °C[9], other membranes do not have sufficient permeance to satisfy practical $CO_2$ separation requirements. In addition, there is always a sharp compromise between permeance and selectivity. In contrast, our ultra-thin enzymatic liquid membrane exhibits a combination of high $CO_2$ permeance (up to 2600 GPU) and high $CO_2/N_2$ selectivity (500–788, see gas chromatography results in Supplementary Fig. 4). To demonstrate the overall utility of our membrane for $CO_2$ separation, we further assessed its ability to perform $CO_2/H_2$ separation using a 43% $H_2$ and 57% $CO_2$ gas mixture maintained at ambient pressure (see Supplementary Fig. 5 for setup). In this case we determined $CO_2/H_2$ separation factors as high as 1500 (Supplementary Fig. 6), while maintaining $CO_2$ permeances in the same range as for $CO_2/N_2$ separations.

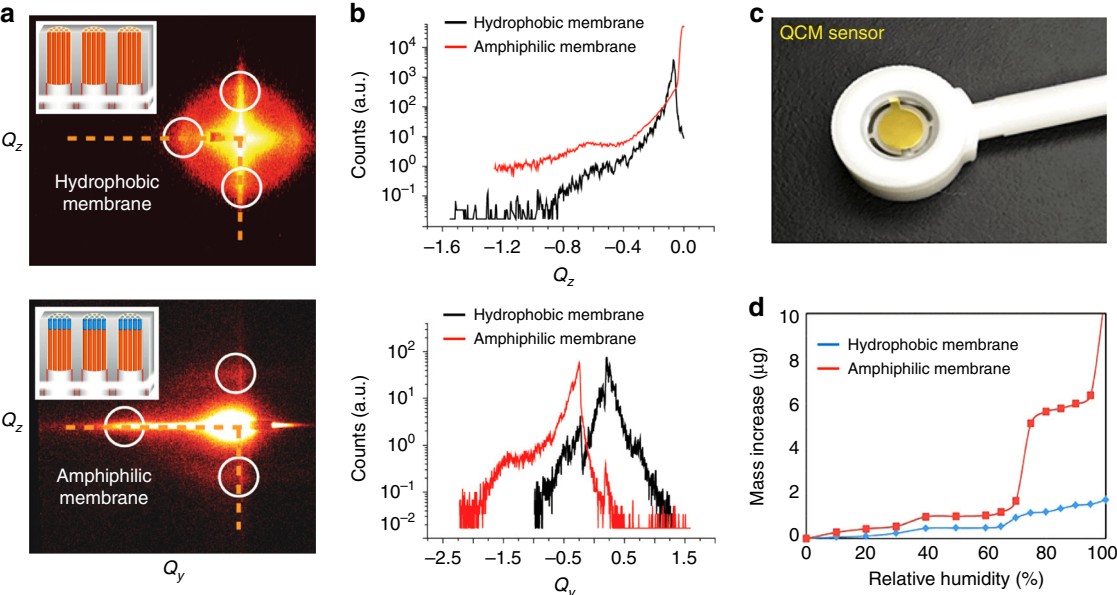

**Fig. 6** GISAXS and water adsorption isotherm characterization of hydrophobic and amphiphilic membranes. **a** GISAXS characterization of the hydrophobic and amphiphilic-surface-modified membrane showing a similar mesoporous structure. **b** Linecuts of GISAXS showing the similarity of two membranes. **c** Photograph of a 25-mm-wide quartz crystal microbalance (QCM) mass sensor coated with a P123-templated mesoporous silica film. **d** QCM mass increase as a function of % relative humidity (RH) for the hydrophobic and amphiphilic membranes showing sudden water adsorption in the amphiphilic membrane (at RH = 75%) due to the capillary condensation of water vapor on the hydrophilic top surface. The mass of condensed water vapor establishes the water volume, from which we calculate the average thickness of the nano-stabilized liquid membrane geometrically assuming that the amphiphilic-surface modified membrane uniformly coats the QCM device

The stability of the membrane was also demonstrated for a period of three months (Supplementary Fig. 7).

In order to explain the enhanced performance of our CA catalytic membrane, we must reconsider the three steps governing the flux and the selectivity of any membrane considered previously: $CO_2$ capture (step I), $HCO_3^-$ transport (step II), and $CO_2$ release (step III). In our case, CA enzymes catalyze the selective and rapid dissolution and regeneration of the target species (steps I and III); short diffusion distances combined with the inherent three orders of magnitude higher diffusivity within liquid vs. polymers commonly used for $CO_2$ membranes[6,14] maximize transport rates (step II). Liquid membranes containing CA have been reported previously for $CO_2$ separation by Ward and Robb in the 1960s[15] and more recently by Carbozyme Inc[27]. However, the inherent mechanical weakness of the water layer in their membrane configurations limited their membrane thicknesses, to be only as thin as 10–100 microns, about a hundred times thicker than most polymer membranes; therefore, negating the potential advantage of the liquid membrane compared to a polymer membrane. Here, through nanoconfinement, we have created a mechanically stable liquid membrane only ~18-nm-thick. Furthermore, compared to the Ward and Robb and Carbozyme membranes another advantage of our membrane is the high enzyme concentration achieved by confinement within the close-packed array of hydrophilic nanopores (see Fig. 4b). CA enzyme solubility in liquid membranes is in general lower than 0.2 mM. For example, Carbozyme was able to use a CA concentration of only 0.16 mM (5 g l$^{-1}$) and Ward and Robb were able to use a CA concentration of only 0.06 mM (2 g l$^{-1}$). In contrast, the high density of hydrophilic nanopores ($3.92 \times 10^{11}$ nanopores per cm$^2$, Supplementary Discussion) in our membrane, if filled with CA, would allow attainment of a significantly higher local CA concentration. To prove this point we performed FTIR spectroscopy of the CA-filled membrane prepared on an IR transparent silicon substrate in the same manner as for the QCM

measurements (Fig. 7e). Based on the molar extinction coefficient of the Amide I absorption band at 1640 cm$^{-1}$ attributed uniquely to the CA enzyme, we calculated a molar concentration of CA that corresponded to a loading of on average of 2 CA enzymes per nanopore yielding an effective CA concentration within the membrane of 3.7 mM (100 mg ml$^{-1}$), or, an effective areal density of $8.0 \times 10^{11}$ CA cm$^{-2}$. This CA concentration is ten times greater than that achievable in solution (~10 mg ml$^{-1}$) and correspondingly accelerates the rates of selective $CO_2$ dissolution and release from the membrane.

The $CO_2$ permeance was then estimated via the calculated CA enzyme areal density. Here we first considered which step, among steps I–III, is rate-limiting for $CO_2$ permeance. Based on the equilibrium described by Eq. 1, CAs catalyze the dissolution of $CO_2$ from the feed side to form bicarbonates that dissociate into carbonic acids and diffuse through the water layer and eventually be enzymatically converted back to $CO_2$ on the 'downstream' side. Hence, the $CO_2$ capture (step I) as well as the $CO_2$ release (step III) are both dependent on the activity of CA enzymes, whereas the $HCO_3^-$ transport (step II) does not and is a function of the diffusion coefficient of carbonate species in water. In regards to the diffusion transport (step II), given the known $CO_2$ permeability in pure water[15] ($210 \times 10^{-9}$ cm$^3$ (STP) cm sec$^{-1}$ cm$^{-2}$ cm$^{-1}$ Hg$^{-1}$), the permeance of the designed 20-nm-thick water-membrane should be of $210 \times 10^{-9}$ cm$^3$ (STP) cm sec$^{-1}$ cm$^{-2}$ cm$^{-1}$ Hg$^{-1}$ divided by the thickness, which is 0.1 cm$^3$ s$^{-1}$ cm$^{-2}$ cm$^{-1}$ Hg$^{-1}$, or $10^5$ GPU. Now, since a $CO_2$ permeance of $10^5$ GPU is much larger than the permeance observed (Fig. 7f), it follows that the CA-catalyzed steps I or III are rate limiting. Correspondingly, we can estimate the flux from the experimentally determined areal density of CA ($8 \times 10^{11}$ molecules cm$^{-2}$) assuming the native enzymatic activity of CA ($10^6$ reactions per second[15]). This results in a calculated $CO_2$ permeance of $8 \times 10^{17}$ molecules sec$^{-1}$ cm$^{-2}$ corresponding to a volumetric flux of 0.03 cm$^3$ sec$^{-1}$ cm$^{-2}$. At the 36 cm Hg driving pressure (see

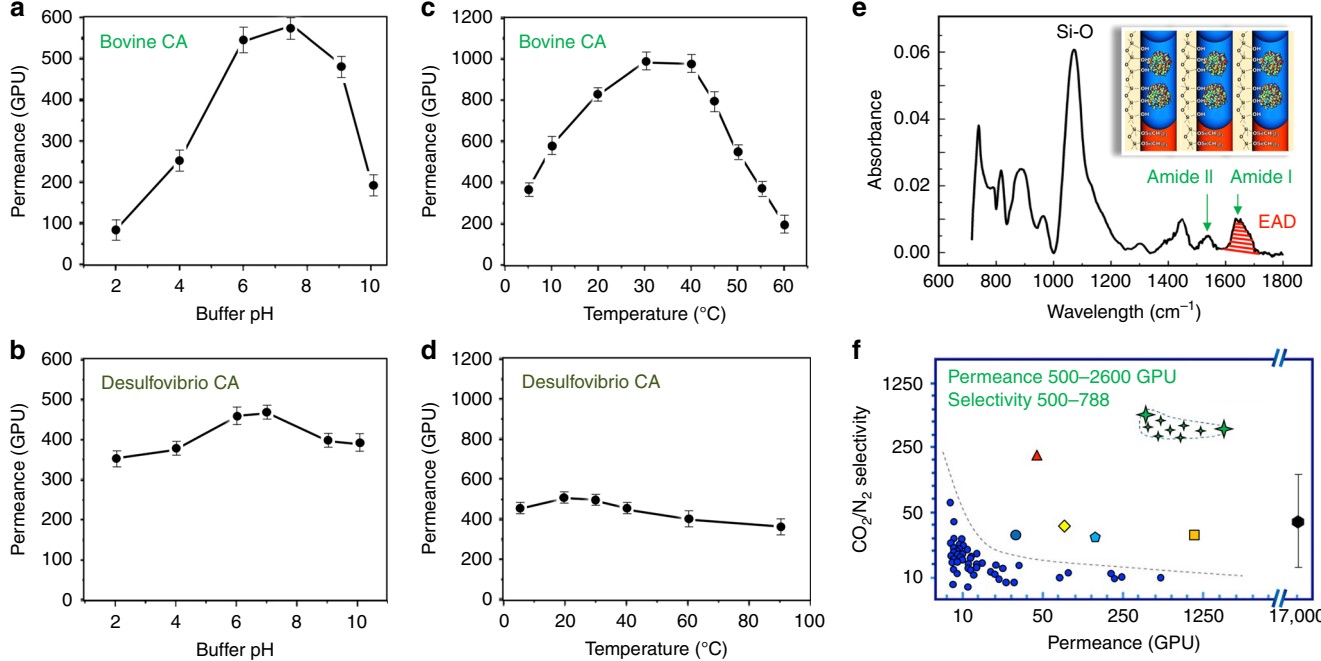

**Fig. 7** $CO_2$ separation applications of the enzymatic liquid membrane. **a**, **b** Comparison of the $CO_2$ permeance of Bovine CA and *Desulfovibrio vulgaris* CA enzymatic liquid membranes as a function of the pH. **c**, **d** Comparison of the $CO_2$ permeance of Bovine CA and *Desulfovibrio vulgaris* CA enzymatic liquid membranes as a function of the temperature. **e** FTIR spectrum of the enzymatic liquid membrane indicating an effective areal density (EAD) of $8.0 \times 10^{11}$ CA cm$^{-2}$ that is consistent with the loading of two CA enzymes per nanopore. **f** Selectivity vs. permeance plot (log 5 scale) of the enzymatic liquid membrane compared with various membranes. The green stars are performance range for our ultra-thin enzymatic liquid membrane; the red triangle is the performance for the Carbozyme CA membrane[19;] the blue dots, the ocean blue circle, the yellow prism, the light blue pentagon, the orange square, and the black hexagon are $CO_2$ membrane performance data respectively from references[7,49–53,10]. The selectivity of the ZSM-5 membranes (black hexagon)[10] increases from 17 to 210 as the temperature decreases from 37 to −43 °C. Error bars represent 95% confidence intervals for experiments performed with $n = 3$

Methods), this corresponds to a $CO_2$ permeance of 833 GPU, which is within the measured data range (500–2600 GPU, Fig. 7f), but considerably lower than the highest permeance measured.

**Molecular dynamics simulations of nanoconfined CA enzymes.** We reasoned that the variability in the measured permeance and the discrepancy between the calculated 'theoretical' value and the highest measured $CO_2$ permeance could be a consequence of nanoconfinement effects that might influence/enhance the enzymatic activity. In order to test this idea, we performed molecular dynamics simulations of the CA enzyme confined within 8 nm diameter silanol-terminated mesopores, under conditions that mimic the operational conditions of the liquid membrane, to characterize atomic-level details of the system. As shown in Fig. 8a, we simulated one or more CA enzymes in a rectangular silica nanopore (inner dimensions of $8 \times 8 \times 10$ nm) that is filled with water at pH 7 (the average silanol density = 5.9 Si–OH nm$^{-2}$ with 16.5% ionization, see Methods).

The simulations revealed that, initially placed in the center of the nanopore, the CA enzymes rapidly (<100 ns) diffuse toward the walls of the pore (Fig. 8b) and form hydrogen bonds that are sustained throughout the simulation (Fig. 8c). Adsorption to the walls of the nanopore is expected due to the large number of polar and charged (positive and negative) residues on the surface of CA, and is consistent with previous studies showing binding of polypeptides to different silica surfaces[28,29]. However, adsorbed CA enzymes retain some mobility and are able to move along the silica surface. Different portions of the enzyme contact the pore walls at different times with the active site remaining accessible to the solution and permitting substrate and product molecules to readily diffuse in and out. The structure of the CA enzyme in the

nanopore is highly robust, as shown by the root-mean-squared deviation (RMSD) of the backbone and active site atoms compared to the CA crystal structure (Fig. 8d–f), and does not appear to be negatively affected by adsorption to the nanopore. Furthermore, the CA RMSD data for simulations in the nanopore closely resemble the values obtained for the free enzyme in solution (Fig. 8f), even for the case of crowded confinement (2–4 CA enzymes in the nanopore with an effective concentration greater than 150 mg ml$^{-1}$ within individual nanopores). These results indicate that the enzymatic activity of CA confined within silica nanopores should not be diminished by adsorption and/or crowding. Further, we cannot rule out the possibility that the (effective) CA enzymatic-specific activity could be increased due to molecular crowding in the nanopores as experimentally observed for other confined enzymes[28,29], or that the effective binding affinity could be enhanced (decreased Michaelis constant) due to excluded volume effects[29,30]. This may explain the generally higher levels of $CO_2$ flux measured experimentally compared to values calculated assuming native enzymatic activities.

**Discussion**
Separation processes in natural biological systems typically take place in an aqueous environment at ambient pressure driven by the prevailing chemical potential gradient. Oftentimes separations are aided by enzyme catalysis, and the thickness of biological membranes is normally on the nanometer scale. To implement these natural design strategies for $CO_2$ capture and separation, we have fabricated an ultra-thin, enzymatic, nano-stabilized liquid membrane. By using nature's design principles of ultra-thin membranes and enzymatic aqueous media, we achieved a combination $CO_2$ flux and $CO_2/N_2$ selectivity under ambient pressure

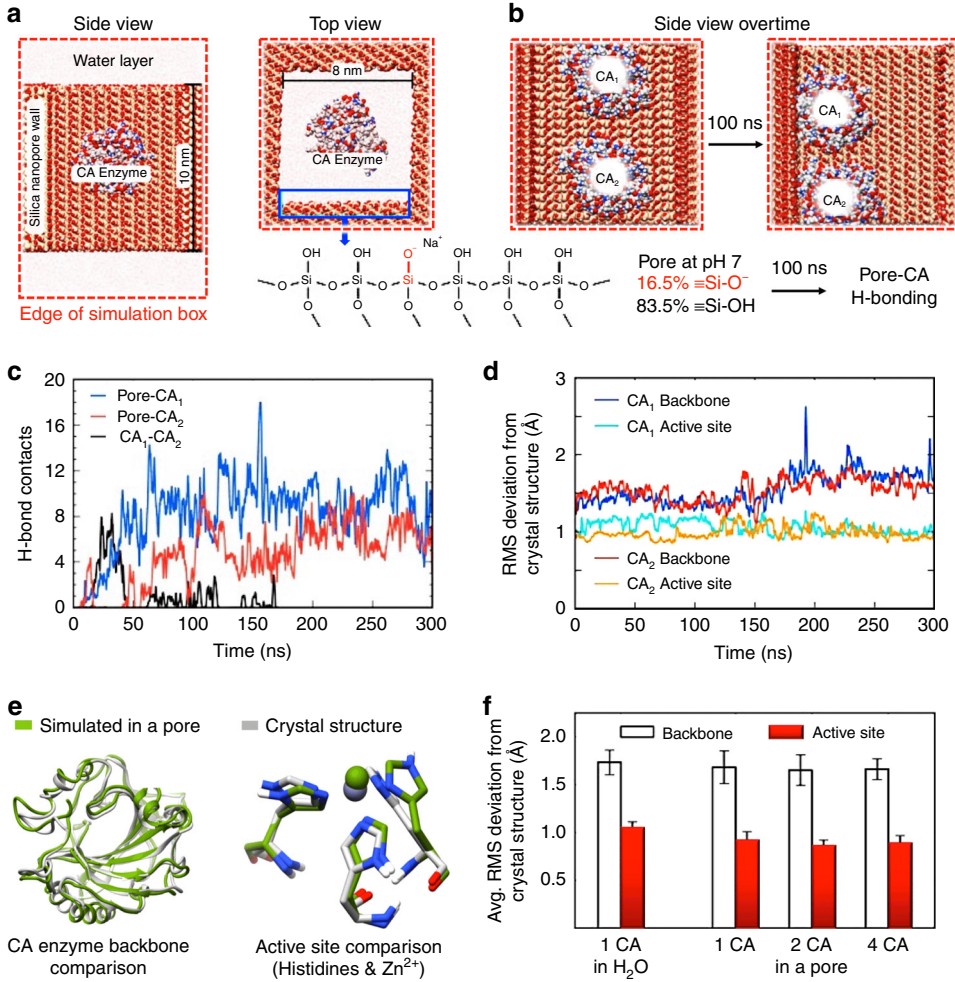

**Fig. 8** Molecular dynamics simulations of carbonic anhydrase confinement within individual mesopore channels of the enzymatic liquid membrane. **a**, **b** Setup for the molecular dynamics simulations of CA enzymes confined within a silica nanopore. **c** CA enzymes rapidly (<100 ns) adsorb to the surface of the silica nanopore and remain in contact with the pore walls for the duration of the simulation, as shown by H-bond contacts between the pore and CA in **c**. **d** Root-mean-squared deviation (RMSD) data of the protein backbone and active site atoms shows that the structure of the CA enzymes adsorbed to the pore remains stable during the course of the simulation and close to the crystal structure. **e** Ribbon representation (left) and close-up of the active site of the CA enzyme show close similarity between the average structure simulated in the nanopore and crystal structure. **f** Average RMSD data (over the last 50 ns of 300 ns simulations) for 1, 2, and 4 CA enzymes per pore (to simulate varying crowded conditions) shows that the structure of the enzyme is highly robust and resembles that of the free CA structure in solution. Error bars represent 95% confidence intervals for experiments performed with $n = 3$

and temperature conditions greatly exceeding that of conventional polymer or inorganic membranes. The membrane design employs a regular array of close-packed 8 nm diameter hydrophilic nanopores whose depth is only 18 nm to confine and stabilize water plus CA enzymes. The high density of CA-filled nanopores establishes an effective CA concentration ten times greater than possible in aqueous solution. At low pressure, the CA enzyme array catalyzes the rapid and selective capture of $CO_2$ via dissolution to form carbonic acid $H_2CO_3$ on the upstream side and the conversion of bicarbonate ($HCO_3^-$) to $CO_2$, which is released on the downstream side. The short water-filled channels minimize diffusional constraints. Altogether, this design maximizes the three steps governing the flux and the selectivity of a $CO_2$ membrane: $CO_2$ capture (step I), $HCO_3^-$ transport (step II), and $CO_2$ release (step III) and enables our enzymatic liquid membrane to exceed Department of Energy standards for $CO_2$ sequestration technologies. Because selectivity is dependent on the exquisite catalytic activity of CA, it should effectively separate $CO_2$ from any gas or gas mixture as we demonstrated for $H_2$/$CO_2$. By simple replacement of CA enzymes with alternate

enzymes, we propose that our ultra-thin, enzymatic, nano-stabilized liquid membrane concept could be readily adapted to other separation processes.

Concerning stability, the membrane is predicted to be mechanically stable because the capillary pressure of water condensed within uniform hydrophilic 8 nm nanopores is ~35 atmospheres. This should prevent water displacement under operations like $CO_2$ capture from flue gas, where the gas pressure is typically less than several atmospheres. The small uniform pore size also confers environmental stability. Based on the Kelvin equation, the membrane should remain water-filled if the RH is maintained above 75%, which is less than that of flue gas streams, which are typically oversaturated in water. For mammalian-derived CA enzymes, the optimal operation temperature would be ca. 37 °C (compatible with flue gas $CO_2$ sequestration), but extremophile enzymes could enable higher temperature operation albeit with less efficiency. Finally, we should consider whether the membrane would be de-activated by impurities known to be present in flue gas. Here, Lu et al.[31] employed CA to promote the adsorption of $CO_2$ from a gas stream containing major flue gas

impurities into a polycarbonate solution and concluded that the concentrations of up to 0.9 mol l$^{-1}$ SO$_2^-$, 0.2 mol l$^{-1}$ NO$_3^-$, and 0.7 mol l$^{-1}$ Cl$^-$, (which exceed the concentrations of typical flue impurities) did not influence the kinetics of absorption from a CA-loaded potassium carbonate solution. Taken together these considerations suggest that the enzymatic liquid membrane is stable enough for use in CO$_2$ capture from flue gas. Additionally, based on its low pressure/temperature performance, it could be considered for other applications like CO$_2$ sequestration in manned space flights.

Concerning cost and scaleability, the unit operations of our membrane synthesis, viz, EISA of ordered mesoporous silica films via dip-coating or spin-coating[32–35], ALD[23,36,37], and plasma processing[23,36] are all scaleable and used today in the micro-electronics industry and in roll-to-roll printing operations (see for example ref. [38]). For demonstration purposes and to compare with other reported membranes, we used a costly commercial 25 mm diameter anodic alumina substrate (Anodisc) for our support. Further, to rigorously control chemistry, we employed calcination to remove surfactant templates and multiple steps of ALD to modify (hydrophobize) the mesoporous silica pore surfaces. To achieve scaleability and reduce costs, the Anodisc could be replaced with tubular alumina supports as employed previously by us for microporous silica membranes[39,40], and by Korelskiy et al. for zeolite membranes[10]. Here it is noteworthy that based on their high flux and selectivity, modules of zeolite membranes prepared on tubular alumina supports were found to be 33% cheaper than a commercial spiral-wound polymer membrane unit for separation of 300 tons of CO$_2$ per day during operation at 10 bars and room temperature. Our membranes have ten times lower flux, but ten times greater selectivity and operate at atmospheric pressure, so similar cost reductions might be expected. However, by replacing calcination with oxygen plasma treatment[22], and ALD with CVD or other large-scale vapor phase methods, it is conceivable that enzymatic liquid membranes could be processed on low cost hollow fiber polymer supports, which would dramatically reduce the cost of CO$_2$ capture technologies.

## Methods

**Materials.** The membranes were fabricated on Whatman© Anodisc porous anodic alumina disc supports purchased from Whatman International Ltd. Bovine CA enzymes were purchased from Sigma-Aldrich, St Louis MO and the *Desulfovibrio vulgaris* CA enzyme was provided from Codexis, Inc.

**Fabrication of ultra-thin enzymatic liquid membranes.** The Whatman© Anodisc porous support is 50-μm-thick and is composed of oriented asymmetric vertical channels that are perpendicular to the disc surface. The channel diameters taper from 200 nm in diameter on the bottom surface to 50–100 nm in diameter on the top surface (see Fig. 3). The support was treated with UV/ozone to fully hydroxylate the alumina surface and insure wetting and covalent bonding with the 'sol-gel derived' silica mesophase (vide infra). In order to fabricate oriented 8 nm diameter cylindrical pores within the channels of the Anodisc, we prepared a Pluronic P123 block copolymer containing silica sol following our reported procedure[20,21]. The sol was applied by spin-coating at 3000 rpm where capillary action followed by EISA[20,21] resulted in the formation of a hexagonal silica/P123 mesophase oriented within the Anodisc pore channel. After two successive spin-coating depositions, the samples were aged at 50 °C for 12 h. To remove the P123 pore template, the samples were calcined at 400 °C for 2 h using a heating rate of 1 °C min$^{-1}$. This resulted in 8 nm diameter cylindrical nanopores aligned within the 50–100 nm pores of the Anodisc as shown in Fig. 3.

To enable the formation of an ultra-thin, stabilized liquid membrane, we first exposed the Anodisc to ozone irradiation to maximize the coverage of hydroxyl groups on all the nanopore surfaces. This was followed by three cycles of alternating HMDS + TMCS/H$_2$O vapor exposure at 180 °C in an Angstrom-dep$^{TM}$ ALD system to convert the hydrophilic surface hydroxyl groups to hydrophobic trimethylsilyl groups. Following that, the hydrophobic porous support was placed into the plasma chamber of an Angstrom-dep$^{TM}$ III plasma-ALD system, and the top surface was irradiated by an oxygen plasma for 5 s, converting only an 18-nm-deep thickness of the hydrophobic nanopores to hydrophilic hydroxyl terminated silica nanopores. In order to load CA enzymes into the nanopore channels, the membrane was 'floated' hydrophilic face down on a 0.05 mM CA solution and bath

sonicated gently for 10 min. Then the samples were removed from the solution, inverted, and maintained in a horizontal configuration on a clean surface until all excess water on the membrane evaporated.

**Structural and physical characterization.** Focused ion beam and scanning electron microscopy (FIB/SEM) experiments were carried out on a FEI Q3D dual beam FIB/SEM system, with 30 kV/3 nA initial voltage/current followed by 8 kV/25 pA final polishing voltage/current for ion beam mode, and 5 kV/24 pA for scanning electron microscopy mode. Transmission electron microscopy images were acquired using a JEOL2010F HRTEM, and Ti-mapping was acquired using the same TEM with a Gatan EELS system. GISAXS was performed using a Bruker Nanostar on samples prepared on Anodisc substrates fabricated as indicated above or on Si substrates prepared as described for Fourier-transform infrared analysis (vide infra). Quartz crystal microbalance analyses were performed using a QCM200-5MHz QCM manufactured by Stanford Research Systems. A home-built, air-tight environmental chamber equipped with gas flow controllers to perform the H$_2$O isotherms. Fourier-transform infrared spectroscopy was performed using a Thermo Scientific Nicolet 6700 Fourier-transform infrared spectrometer. A P123-templated silica film was deposited onto intrinsic, IR transparent single crystal Si substrates (400-μm-thick, double-polished) by spin-coating; this film was then processed in an identical manner as the Anodisc supported P123-templated film described above and loaded with CA.

**CO$_2$ separation performance measurement.** CO$_2$ permeance and CO$_2$/N$_2$ or CO$_2$/H$_2$ selectivity measurements were performed using a home-made test cell designed to accommodate a 25 mm diameter sample and to be immerged into a water bath for needed temperature control. The feed gas was first introduced through a water bubbler heated at 90 °C to achieve a saturated humidity. In the permeance vs. temperature and pH measurements (Fig. 7a, b), the feed gas was compressed pure CO$_2$ with a relative pressure of 36 cm Hg or 0.48 bar. Control experiments of CO$_2$ or Ar permeance were performed using liquid membranes prepared without CA, and CO$_2$ and Ar were found to be undetectable using a bubble flowmeter. For the CO$_2$/H$_2$ separation procedure (See Supplementary Fig. 5), gas membranes were delivered in a sealed stainless-steel vessel, and used as is without further modification. A cross-flow configuration was used for H$_2$ permeation measurements. Feed gas composition was fixed at 43% H$_2$ and 57% CO$_2$. The quantity of gas permeating across the membrane was calculated by the difference in gas flow at the inlet vs. the exhaust, with a typical cross-flow rate of 0.21 ccm. Gas permeated across the membrane was then carried by an Ar gas (8.01 ccm) into a calibrated Inficon 3000 Micro GC gas analyzer for quantitative measurement discrimination.

**Molecular dynamics simulations.** The simulations were performed with the GROMACS software package[27]. The CHARMM36 force field[41,42] was used to model the bovine CA enzyme (Protein Data Bank accession number 1V9E[43]) under different conditions relevant to CO$_2$ separation, including interaction with silica nanopores. The silica nanopore atoms were modeled with the CHARMM36-compatible INTERFACE force field[44,45]. Protonation states of amino acids of the CA enzyme were selected according to the results of PROPKA analysis at pH 7[46]. A rectangular silica nanopore was built based on the structure of the alpha-cristobalite unit cell. The pore's outer dimensions are 11 × 12 × 10 nm and its internal dimensions are 8 × 8 × 10 nm. The average surface silanol density of the pore is 5.9 Si–OH nm$^{-2}$, which provides a reasonable model of the amorphous silica surface used in the experimental membranes[47,48]. A percentage (16.5%) of the surface silanols were ionized to match the pH 7 conditions. Sodium (Na$^+$) ions were added to counter the negative charge of the ionized silanol groups. No additional salt molecules, either Na$^+$ or Cl$^-$, were added to the simulation, except to produce an overall neutral charge simulation system. A vertical water-filled space exists between periodic images of the simulation cell of height 6 nm, giving the CA enzyme the ability to exit the nanopore. Three CA-nanopore systems were simulated with one, two, and four enzymes within the pore to observe possible crowding effects. A free CA enzyme in solution was also simulated for reference. All systems were simulated at room temperature (298 K) for 300 ns using a Nose–Hoover thermostat. The simulation volume for pore systems was adjusted during the early stages of the simulation to obtain an average pressure of 1 atm, and subsequently simulated at constant volume. The free CA enzyme was simulated at constant 1 atm pressure using a Parrinello–Rahman barostat.

**Data availability**. All relevant data are available from the authors on request.

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

## Acknowledgements

C.J.B., S.B.R., J.M.V., E.C., and S.C. acknowledge support by the Sandia National Laboratory Laboratory-Directed Research and Development Program. C.J.B., Y.-B.J., and Y.F. acknowledge support from the US Department of Energy, Office of Science, Division of Catalysis Science under Grant No. DE-FG02-02ER15368 and the Air Force Office of Scientific Research under Grant No. FA9550-14-1-0066. C.J.B. also acknowledges support from the Department of Energy Office of Science, Division of Materials Science and Engineering. This work was supported, in part, by the National Science Foundation under Cooperative Agreement No. EEC-1647722. Any opinions, findings, and conclusions or recommendations expressed in this material are those of the author(s) and do not necessarily reflect the views of the NSF. This work was performed, in part, at the Center for Integrated Nanotechnologies (CINT), an Office of Science User Facility

operated for the U.S. DOE's Office of Science by Los Alamos National Laboratory (Contract DE-AC52 06NA25296). Sandia National Laboratories (SNL) is a multi-mission laboratory managed and operated by National Technology and Engineering Solutions of Sandia, LLC, a wholly owned subsidiary of Honey-well International, Inc., for the U.S. Department of Energy's National Nuclear Security Administration under contract DE-NA-0003525.

## Author contributions

Y.-B.J., S.B.R., J.L.C., and C.J.B. invented the enzymatic liquid membrane concept. Y.F. fabricated the membrane in the laboratories of C.J.B. and Y.-B.J. with the help of H.X.Z. who performed the atomic layer deposition. Y.F., S.C., and Y.-B.J. designed and executed the gas permeation experiments. D.D., H.F.X., and E.C. performed physicochemical characterization studies. J.M.V. designed and executed the modeling study in collaboration with S.B.R. C.J.B. wrote the manuscript, with contributions from S.B.R.; J.G.C. and C.J.B. revised the manuscript.

## Additional information

**Competing interests:** The authors declare no competing interests.

