## [Peer Review File · Nature Communications]

Reviewers' comments:

Reviewer #1 (Remarks to the Author):

The submitted work appears of great interest and value for the scientific Community, and is therefore recommended for publication in Nature after minor revision based on the comments below:

1. In the introduction, the authors are making a bold statement about other membrane materials, saying that most current membrane materials display poor performance for CO₂ separation. This is only true in part. There have been examples of both porous and dense membranes exhibiting a good combination of permeance and selectivity, and this has to be acknowledged and addressed in the work. For instance, Korelskiy et al. (J. Mater. Chem. A, 2015, 3, 12500–12506) reported a CO₂/H₂ selectivity of ca. 200 with a permeance of ca. 17000 GPU.

2. It appears that only CO₂/N₂ separation was studied in the work. It is therefore unfair to call the paper "... for CO₂ separation". Different CO₂/gas mixtures have a very different behaviour, and these membranes may be totally useless for CO₂/H₂ separation carried out at elevated pressures. Thus, the title of the work and introduction need to be changed to properly address the scope of this work.

3. The authors seem to ignore describing (or studying) the pressure dependence of the permeance. The pressure is a very important factor in gas separation, and some clarification is therefore needed.

4. How practical is the membrane preparation? In other words, is there a potential for scale-upability? Please, comment on that.

Reviewer #2 (Remarks to the Author):

Ultra-thin, enzymatic, nano-stabilized water membrane for CO₂ separation
Yaqin Fu, Ying-Bing Jiang* Darren Dunphy, Haifeng Xiong, Eric Coker, Stan Chou, Hongxia Zhang, Juan M. Vanegas, Joseph L. Cecchi, Susan B. Rempe, C. Jeffrey Brinker*
Nature Materials:

General comments:

The authors report an ultrathin-skinned (18 nm) liquid membrane selective layer for CO₂ separation. The selective layer incorporates a metalloenzyme carbonic anhydrase (CA), water layer stabilized by capillary condensation within a mesoporous silica film, which catalyzes the rapid inter-conversion of CO₂ + H₂O to carbonic acid. This is an unusual class of facilitated transport membrane incorporating CA, which completely relies on maintaining hydration in the membrane (i.e. feed gases). The ultra-thin selective enzymatic membrane is able to separate CO₂ at room temperature with a high permeance of 2600 GPU and very high CO₂/N₂ and CO₂/H₂ mixed-gas selectivities. The reviewer has no arguments about the quality of the work and the obtained results, which may have the novelty necessary for NC, although in the reviewer's opinion, the quality of the Figures is far below the necessary requirement for clear visual communication. While the results suggest these membranes may have among the highest performance, nominally exceeding the DOE performance requirements for CO₂ capture for flue gas (high permeance and gas selectivity), there are several fairly major concerns, outlined below.

Specific comments:

(1) It is unclear about the membrane area that has been studied; is it 1 cm², 10 cm², 50 cm²? Since thousands of square meters are needed for CO₂ capture, how scalable is the membrane fabrication process?

(2) The membrane fabrication procedure seems very complex and multi-step, relying on specialized techniques such as Atomic Layer Deposition (ALD) done in vacuo. Again, how practical is this for scale up? It might be alright on a small scale using alumina membranes, but the membrane fabrication process is multi-step, and large areas are needed.

(3) It is unclear whether this membrane would be robust enough to operate in a flue gas environment, where the temperature would very likely be above room temperature (50-80 C) and where gaseous impurities might poison the catalyst, rendering the CA mode of transport inoperative. Would trace impurities such as NO_x and SO_x have an influence on the metalloenzyme?

(4) If there is variability of moisture in the gas feed, dehydration would also render the CA mechanism inoperative. The authors recognize this to some extent on page 6 "One conceivable concern might be how to ensure that the water membrane is stable and will not 'dry out' in real world applications. As previously discussed, this concern will be alleviated by maintaining the membrane at a sufficient relative humidity where, due to capillary condensation, the uniformly sized hydrophilic nanopores remain water-filled." The authors state "the Kelvin equation predicts condensation to occur at a RH equal to or exceeding 80%". Presumably, if the moisture content falls below this level, the CA mechanism won't have sufficient water to operate. The authors also state "A typical flue gas comprises 6.2wt% H₂O if it is from a coal-fired plant, and 14.6wt% H₂O if from a gas-fired plant. Both are much higher than the saturated water vapor concentration at 40°C (~50g H₂O/kg air or 0.5wt% H₂O)." However, the reviewer thought that coal-fired power flue gas is higher than 40 C, and more typically 60 C.

(5) In general, the Figures are too cluttered and too small to comfortably see what the work is about. Even magnified on a screen, some of the images are simply too small, making it tedious and tiring to work one's way through this work. Good Figures are essential in bringing clarity to any high level work, and this is not the case in this paper. Most of the Figures will need to be reworked, and some of them will need to remove some of the panes and place them in the ESI section. For example: Scheme 1, both images are too small; the left image one cannot easily see the molecules, and on the right pane, the writing is too small. Scheme 2, the top two images are far too small, and the writing on the lower part is much too small. It makes it tedious or hard to figure out what is going on. Figure 1, left pane, very poor choice of color for the oval showing where the membrane surface is sampled from; right pane is a confusing mixture of TEM images. Figure 3d, writing is too small. Figure 4 most of it is illegible, and too hard to try to read it or figure out what is going on.

(6) A recent review on high-permeability membrane materials for CO₂ separations was published by a Chinese group, which would be relevant to include in the bibliography (Energy Environ. Sci., 2016, 9, 1863).

(7) On page 3, the authors mention "nano-stabilized liquid membrane", but this term is not as clear to the reader as it could be, unless they mean the water is stabilized by capillary action. More explanation is needed.

(8) Figure 1(c): Is this actually cross-sectional or surface view? If cross-sectional, it suggests a pore structure along the plane of the membrane. In the Methods section, it states "The membrane was fabricated on a commercial, asymmetric Anodisc porous alumina support purchased from Whatman International Ltd. The support is 60µm thick and is composed of oriented asymmetric

vertical channels that are perpendicular to the disc surface." What is the active membrane area made by this multi-step process?

(9) Typo on page 12: "converting only an 18-nm deep thickness"

(10) The authors report data for gas mixtures CO₂/N₂ and CO₂/H₂. While CO₂ capture from flue gas (CO₂/N₂) would supply feed gas marginally above atmospheric pressure, what about the CO₂/H₂ mixture? Any higher pressures would push out the water layer, rendering the CA mechanism inoperable.

(11) Another concern the authors recognize is gas feed pressure: "Another potential concern is that of the water membrane strength, e.g. will the water membrane be ruptured when applying pressurized gas for separation?" This might be very simple to measure (if they didn't already measure it – I cannot make out the graphs very well).

(12) Overall, clarity of Figures and clarity of narrative and explanations have room for improvement.

Reviewer #3 (Remarks to the Author):

Carbonic anhydrases have been widely used to prepare the facilitated transport membranes for CO₂ separation and capture. However, this is not really practical application yet, because of its instability and high cost. In this work, the membrane was fabricated on a commercial, asymmetric porous alumina support, which is higher cost and not easy scale-up, compared with polymeric supports. Therefore, this manuscript is not novel enough to be published in Nature Communications.

1. The enzymatic activity is affected by buffer pH and temperature, the relation of enzymatic activity and permeance could be investigated. The catalytic mechanism of CA for CO₂ should be also studied by enzyme kinetics.
2. Long-lasting stability is very important for the practical application of the membranes, so the long-term stability of the membranes should be tested. These factors should be considered, including water release in the nanopore and enzyme deactivation.
3. In Figure 4, the orders of Figure a, b are not corresponding to the manuscript.
4. How to control and obtain the actual amount of CA enzyme incorporated in each sample?
5. The authors should run a controlled experiment without CA enzymes. The membrane module and set-up for separation performance should be also provided for better understanding.
6. The introduction section is lengthy, and the authors should come to the point of enzyme based membranes.

Response to Reviewers' comments:

We have carefully considered all the reviewer comments and have revised the manuscript and included results of additional substantiating experiments to address their concerns and support the novelty and utility of our enzymatic liquid membrane. To aid in the reviewing process, we have replied to all reviewer comments on a point-by-point basis and highlighted major revised sections in the main text and the SI. We addressed all formatting issues and largely revised all figures to increase their clarity. Overall, I feel the manuscript has been greatly improved by these revisions and we thank the reviewers for their comments, criticisms and suggestions. We hope the manuscript now meets the high standards for publication in *Nature Communications*.

Reviewer #1 (Remarks to the Author):

The submitted work appears of great interest and value for the scientific Community, and is therefore recommended for publication in Nature after minor revision based on the comments below:

Response: We thank this reviewer their careful consideration of our MS and for providing very useful criticisms, suggestions and recommendations. We have revised the MS accordingly and included additional supplementary figures and calculations to clarify points of concern.

1. There have been examples of both porous and dense membranes exhibiting a good combination of permeance and selectivity, and this has to be acknowledged and addressed in the work. For instance, Korelskiy at al. (*J. Mater. Chem. A*, 2015, 3, 12500–12506) reported a CO₂/H₂ selectivity of ca. 200 with a permeance of ca. 17000 GPU.

Response: We thank the reviewer for informing us of this exceptionally relevant recent paper describing CO₂/H₂ separation via a 0.5- μ m thick ZSM-5 zeolite membrane operated at a pressure of 9 bars (incorporated in the revised MS as reference 10, see also reference 11). As we understand it, the selectivity mechanism requires operation of the zeolite membrane at high pressure (close to the triple point of CO₂) where CO₂ is a liquid or will condense as a liquid into the ZSM5 nanopores, via capillary condensation, excluding H₂, which remains in the vapor phase. As we now emphasize, our 'biomimetic' membrane operates near ambient pressure and temperature according to the prevailing chemical potential gradient. Selectivity is achieved by the exceptional activity of CA, which reversibly catalyzes the dissolution of CO₂ into water as carbonic acid on the feed side and carbonic acid to CO₂ on the downstream side. High CO₂ flux is accomplished by the very high effective concentration of CAs within the close packed nanopore array and the 'ultra-thinness' of the membrane (~18-nm). As such we feel that our membrane characteristics and performance are complimentary to those of zeolitic membranes. These points are now made in the revised manuscript.

2. It appears that only CO₂/N₂ separation was studied in the work. It is therefore unfair to call the paper "... for CO₂ separation". Different CO₂/gas mixtures have a very different behaviour, and these membranes may be totally useless for CO₂/H₂ separation carried out at elevated pressures. Thus, the title of the work and introduction need to be changed to properly address the scope of this work.

Response: Agreed – We now include in the Supplementary Information results for H₂/CO₂ separation (Supplementary Figure S5-6), where we show similarly quite high CO₂/H₂ selectivity (as high as 1500) stemming from the exquisite enzymatic selectivity and activity of CA for reversible dissolution of CO₂. The title has been revised to reflect now operation under near ambient pressure conditions.

3. The authors seem to ignore describing (or studying) the pressure dependence of the permeance.

Response: As commented upon in Response 1 our membrane was designed to operate under essentially ambient conditions according to the prevailing chemical potential gradient. We do stipulate that due to stabilization of the confined water within 8 nm pores the membrane could be operated up to 35 bars without destabilization (Supplementary Calculations 1), but we have not explored the overall pressure dependence as the intention was to operate at near ambient pressure conditions.

4. How practical is the membrane preparation? In other words, is there a potential for scale-upability? Please, comment on that.

Response: Yes, good point, which has been made by the other reviewers. Concerning cost and scaleability, the unit operations of our membrane synthesis, viz, evaporation-induced self-assembly of ordered mesoporous silica films via dip- or spin-coating, atomic layer deposition (ALD), and plasma processing (see added refs. 37-45) are all scaleable and used today in the microelectronics industry and in roll-to-roll printing operations (see for example ref. 43). For demonstration purposes and to compare with other reported membranes, we used a costly commercial 25-mm diameter anodic alumina substrate (Anodisc) for our support. Further, to rigorously control chemistry, we employed calcination to remove surfactant templates and multiple steps of ALD to modify (hydrophobize) the mesoporous silica pore surfaces. To achieve scaleability and reduce costs, the Anodisc could be replaced with tubular alumina supports as employed previously by us (refs. 44-45) for microporous silica membranes, and by Korelskiy et al. (ref. 10) for zeolite membranes. Here it is noteworthy that based on their high flux and selectivity, modules of zeolite membranes prepared on tubular alumina supports were found to be 33% cheaper than a commercial spiral-wound polymer membrane unit for separation of 300 tons of CO₂ per day during operation at 10 bar and room temperature. Our membranes have 10x lower flux but 10x greater selectivity and operate at atmospheric pressure, so similar cost reductions might be expected. However, by replacing calcination with oxygen plasma treatment (ref. 24), and ALD with CVD or other large-scale vapor phase methods, it is conceivable that enzymatic liquid

membranes could be processed on low cost hollow fiber polymer supports, which would greatly reduce the cost of CO₂ capture technologies. The cost and scaling aspects are now addressed in the Discussion and Summary Section.

Reviewer #2 (Remarks to the Author):

General comments:

The authors report an ultrathin-skinned (18 nm) liquid membrane selective layer for CO₂ separation. The selective layer incorporates a metalloenzyme carbonic anhydrase (CA), water layer stabilized by capillary condensation within a mesoporous silica film, which catalyzes the rapid inter-conversion of CO₂ + H₂O to carbonic acid. This is an unusual class of facilitated transport membrane incorporating CA, which completely relies on maintaining hydration in the membrane (i.e. feed gases). The ultra-thin selective enzymatic membrane is able to separate CO₂ at room temperature with a high permeance of 2600 GPU and very high CO₂/N₂ and CO₂/H₂ mixed-gas selectivities. The reviewer has no arguments about the quality of the work and the obtained results, which may have the novelty necessary for NC, although in the reviewer's opinion, the quality of the Figures is far below the necessary requirement for clear visual communication. While the results suggest these membranes may have among the highest performance, nominally exceeding the DOE performance requirements for CO₂ capture for flue gas (high permeance and gas selectivity), there are several fairly major concerns, outlined below.

Response: We thank this reviewer their careful consideration of our MS and for providing very useful criticisms, suggestions, and recommendations. We agree and apologize about the poor quality of the figures included in the original manuscript and have taken every effort to improve them in the revised manuscript. We have also included additional supplementary figures and calculations to clarify points of concern.

Specific comments:

1. It is unclear about the membrane area that has been studied; is it 1 cm², 10 cm², 50 cm²? Since thousands of square meters are needed for CO₂ capture, how scalable is the membrane fabrication process?

Response: Good point made by all reviewers. For demonstration purposes and to be able to make comparisons with some other reported membranes, we employed a 6.5 cm² commercial and costly Anodisc porous support. Concerning scalability, as noted above in response to Response to Reviewer 1:

Concerning cost and scalability, the unit operations of our membrane synthesis, viz, evaporation-induced self-assembly of ordered mesoporous silica films via dip- or spin-coating, atomic layer deposition (ALD), and plasma processing (refs. 37-45 were added) are all scalable and used today in the microelectronics industry and in roll-to-roll

printing operations (see for example ref. 43). For demonstration purposes and to compare with other reported membranes, we used a costly commercial 25-mm diameter anodic alumina substrate (Anodisc) for our support. Further, to rigorously control chemistry, we employed calcination to remove surfactant templates and multiple steps of ALD to modify (hydrophobize) the mesoporous silica pore surfaces. To achieve scaleability and reduce costs, the Anodisc could be replaced with tubular alumina supports as employed previously by us (refs. 44-45) for microporous silica membranes, and by Korelskiy et al. (ref. 10-11) for zeolite membranes. Here it is noteworthy that based on their high flux and selectivity, modules of zeolite membranes prepared on tubular alumina supports were found to be 33% cheaper than a commercial spiral-wound polymer membrane unit for separation of 300 tons of CO₂ per day during operation at 10 bar and room temperature. Our membranes have 10x lower flux but 10x greater selectivity and operate at atmospheric pressure, so similar cost reductions might be expected. However, by replacing calcination with oxygen plasma treatment (ref. 24), and ALD with CVD or other large-scale vapor phase methods, it is conceivable that enzymatic liquid membranes could be processed on low cost hollow fiber polymer supports, which would greatly reduce the cost CO₂ capture technologies. The cost and scaling aspects are now addressed in the Discussion and Summary Section.

2. The membrane fabrication procedure seems very complex and multi-step, relying on specialized techniques such as Atomic Layer Deposition (ALD) done in vacuo. Again, how practical is this for scale up? It might be alright on a small scale using alumina membranes, but the membrane fabrication process is multi-step, and large areas are needed.

Response: Agreed, we have addressed this concern in the previous response.

3. It is unclear whether this membrane would be robust enough to operate in a flue gas environment, where the temperature would very likely be above room temperature (50-80 C) and where gaseous impurities might poison the catalyst, rendering the CA mode of transport inoperative. Would trace impurities such as NO_x and SO_x have an influence on the metalloenzyme?

Response: Agreed this is an important point – here it should be emphasized that CA enzymes are currently under consideration for CO₂ scrubbers where the influence of possible impurities on the catalytic activity has been quantified. Here Bond *et al.* (ref. 30) show that there is little indication of inhibition of catalytic activity at impurity concentrations that might be expected in such applications (flue gas scrubbing) (i.e. below 100–200 mmol L⁻¹ of SO₄²⁻ or NO₃⁻). Similarly, Lu *et al.* (added ref. 31) employed CA to promote the adsorption of CO₂ from a gas stream containing major flue gas impurities into a polycarbonate solution and concluded that: “concentrations of up to 0.9 mol L⁻¹ SO₄²⁻, 0.2 mol L⁻¹ NO₃⁻ and 0.7 mol L⁻¹ Cl⁻ did not influence the kinetics of absorption from a CA loaded potassium carbonate solution”. With respect to temperature, we demonstrated the use of an extremophile carbonic anhydrase, *Desulfovibrio* CA, to enable operation to 90°C (Fig. 5c) albeit at somewhat lower

efficiency. Overall in our interactions with industry we believe that there is a rather broad operational space in which to perform CO₂ separation that would allow operation under temperature and relative humidity conditions where our membrane is stable. Issues surrounding impurity effects and temperature of operation are now discussed in the Discussion and Summary Section.

4. If there is variability of moisture in the gas feed, dehydration would also render the CA mechanism inoperative. The authors recognize this to some extent on page 6 “One conceivable concern might be how to ensure that the water membrane is stable and will not ‘dry out’ in real world applications. As previously discussed, this concern will be alleviated by maintaining the membrane at a sufficient relative humidity where, due to capillary condensation, the uniformly sized hydrophilic nanopores remain water-filled.” The authors state “the Kelvin equation predicts condensation to occur at a RH equal to or exceeding 80%”. Presumably, if the moisture content falls below this level, the CA mechanism won’t have sufficient water to operate. The authors also state “A typical flue gas comprises 6.2wt% H₂O if it is from a coal-fired plant, and 14.6wt% H₂O if from a gas-fired plant. Both are much higher than the saturated water vapor concentration at 40 °C (~50g H₂O/kg air or 0.5wt% H₂O).” However, the reviewer thought that coal-fired power flue gas is higher than 40 C, and more typically 60 °C.

Response: Agreed, the membrane requires hydration of the pores which occurs in empty pores at ~75% RH (Supplementary Calculations 2), whereas flue gas has 10-20X the saturated water vapor concentration at 40 °C depending upon source. This would seem to us to be an acceptable safety margin. Increasing the temperature from 40 to 60 °C increases the saturation pressure by less than 50% from 8 to 11 KPa, and the membrane should remain fully water saturated. Also, we report that extremophile CA enzymes could allow operation at temperatures exceeding 60 °C. Issues surrounding temperature effects are now discussed in the Discussion and Summary Section.

5. In general, the Figures are too cluttered and too small to comfortably see what the work is about. Even magnified on a screen, some of the images are simply too small, making it tedious and tiring to work one’s way through this work. Good Figures are essential in bringing clarity to any high-level work, and this is not the case in this paper. Most of the Figures will need to be reworked, and some of them will need to remove some of the panes and place them in the ESI section. For example: Scheme 1, both images are too small; the left image one cannot easily see the molecules, and on the right pane, the writing is too small. Scheme 2, the top two images are far too small, and the writing on the lower part is much too small. It makes it tedious or hard to figure out what is going on. Figure 1, left pane, very poor choice of color for the oval showing where the membrane surface is sampled from; right pane is a confusing mixture of TEM images. Figure 3d, writing is too small. Figure 4 most of it is illegible, and too hard to try to read it or figure out what is going on.

Response: We agree and apologize about the quality of the original figures and now have largely replaced all of them with revised figures that we believe to help greatly in illustrating our points.

6. A recent review on high-permeability membrane materials for CO₂ separations was published by a Chinese group, which would be relevant to include in the bibliography (Energy Environ. Sci., 2016, 9, 1863).

Response: Yes, agreed, we now cite this reference which is included as ref. 9.

7. On page 3, the authors mention “nano-stabilized liquid membrane”, but this term is not as clear to the reader as it could be, unless they mean the water is stabilized by capillary action. More explanation is needed.

Response: Yes, that is exactly what we mean. Now in a following sentence in the same paragraph, we define nano-stabilized as follows: “Through capillary condensation, the pores are filled with water plus carbonic anhydrase enzymes confined and stabilized to high pressures by nano-confinement (approximately the capillary pressure, ~35 atmospheres, exerted by water within a hydrophilic 8-nm diameter nanopore).” (See Supplementary Calculations S1)

8. Figure 1(c): Is this actually cross-sectional or surface view? If cross-sectional, it suggests a pore structure along the plane of the membrane. In the Methods section, it states “The membrane was fabricated on a commercial, asymmetric Anodisc porous alumina support purchased from Whatman International Ltd. The support is 60µm thick and is composed of oriented asymmetric vertical channels that are perpendicular to the disc surface.” What is the active membrane area made by this multi-step process?

Response: Sorry for any confusion. The original Fig. 1(c) is a plan view of the mesoporous silica nanopore array at the top surface of the membrane, showing close packed ~8 nm diameter pores that are used to house the CA enzymes (see current Fig. 1f) The effective membrane area is the total pore area of the Anodisc terminating at the Anodisc surface (where the Anodisc pores are completely filled with arrays of 8-nm diameter hydrophilic silica mesopores). The inset in Fig. 1b shows a Focused-Ion Beam prepared plan-view section of the Anodisc surface where the alumina becomes etched and we see the silica mesopore arrays that were contained within and completely fill the Anodisc pores (Note if the pores are at all mis-aligned with the TEM viewing axis they appear as stripe patterns). Supplementary Fig. S2 shows a SEM images of the Anodisc top surface. From analysis of these images we determined the effective pore area to be 70% of the geometric Anodisc area. The Anodisc pores are filled with arrays of mesopores resulting in a density of hydrophilic mesopores of 3.92×10^{11} nanopores per cm² (see Supplementary calculations 3). This is consistent with our experimentally determined enzyme concentration where all silica mesopores, which reside exclusively in the active area of the Anodisc, contain on average 2 enzymes per mesopore. We now clarify these points in Fig. 1 and Supplementary Fig. S2 captions.

9. Typo on page 12: “converting only am 18-nm deep thickness”

Response: Corrected

9. The authors report data for gas mixtures CO₂/N₂ and CO₂/H₂. While CO₂ capture from flue gas (CO₂/N₂) would supply feed gas marginally above atmospheric pressure, what about the CO₂/H₂ mixture? Any higher pressures would push out the water layer, rendering the CA mechanism inoperable.

Response: To address this concern, we state in our summary at the end of the Introductory Section,

“Through capillary condensation, the pores are filled with water plus carbonic anhydrase enzymes confined and stabilized to high pressures by nano-confinement (approximately the capillary pressure, ~35 atmospheres, exerted by water within a hydrophilic 18 nm diameter nanopore).” Please see Supplementary Calculation S1.

Later at the end of the “Formation of sub-20 nm thick enzymatic membrane reactor” Section we state,

“Another potential concern is that of the water membrane strength, e.g. will the water membrane be ruptured when applying pressurized gas for separation? Here, the uniform nano-sized dimensions of the hydrophilic pores assure mechanical stability: the capillary force of water embedded within a small pore can be calculated according to $P = 2\gamma\cos\theta/r$, and for water confined within 8 nm diameter hydrophilic pores where the contact angle θ equals zero, the capillary force is about 35 atm. Therefore, at regular operations, where the gas pressure is typically less than several atmospheres, the capillary force is large enough to stabilize the membrane and prevent its displacement into the hydrophobic portion of the membrane nanopores”.

The point is that if the membrane is operated at ~75% or higher RH, the nanoconfined water plus CA enzymes should not be displaced/pushed out unless very high pressures are applied Please see Supplementary Calculation S2.

10. Another concern the authors recognize is gas feed pressure: “Another potential concern is that of the water membrane strength, e.g. will the water membrane be ruptured when applying pressurized gas for separation?”

Response: As stated in the previous response the membrane when maintained and operated at 75% or higher RH should remain stable to high pressure, whereas it operates at low/zero differential pressure, driven by the prevailing chemical potential gradient. We haven’t attempted to pressurize it to the maximum tolerated pressure, but as we understand flue gas is not highly pressurized - rather it is near ambient pressure.

Reviewer #3 (Remarks to the Author):

1. Carbonic anhydrases have been widely used to prepare the facilitated transport membranes for CO₂ separation and capture. However, this is not really practical application yet, because of its instability and high cost. In this work, the membrane was fabricated on a commercial, asymmetric porous alumina support, which is higher cost and not easy scale-up, compared with polymeric supports. Therefore, this manuscript is not novel enough to be published in Nature Communications.

Response: Yes, good point, which has been made by the other reviewers. Concerning cost and scalability, the unit operations of our membrane synthesis, *viz.*, evaporation-induced self-assembly of ordered mesoporous silica films via dip- or spin-coating, atomic layer deposition (ALD), and plasma processing (refs. 37-45 were added) are all scaleable and used today in the microelectronics industry and in roll-to-roll printing operations (see for example ref. 43). For demonstration purposes and to compare with other reported membranes, we used a costly commercial 25-mm diameter anodic alumina substrate (Anodisc) for our support. Further, to rigorously control chemistry, we employed calcination to remove surfactant templates and multiple steps of ALD to modify (hydrophobize) the mesoporous silica pore surfaces. To achieve scalability and reduce costs, the Anodisc could be replaced with tubular alumina supports as employed previously by us (refs. 44-45) for microporous silica membranes, and by Korelskiy et al. (ref. 10-11) for zeolite membranes. Here it is noteworthy that based on their high flux and selectivity, modules of zeolite membranes prepared on tubular alumina supports were found to be 33% cheaper than a commercial spiral-wound polymer membrane unit for separation of 300 tons of CO₂ per day during operation at 10 bar and room temperature. Our membranes have 10x lower flux but 10x greater selectivity and operate at atmospheric pressure, so similar cost reductions might be expected. However, by replacing calcination with oxygen plasma treatment (ref. 24), and ALD with CVD or other large-scale vapor phase methods, it is conceivable that enzymatic liquid membranes could be processed on low cost hollow fiber polymer supports, which would greatly reduce the cost of CO₂ capture technologies. The cost and scaling aspects are now addressed in the Discussion and Summary Section. Novelty is the thinness of the membrane and its ability to stabilize CA enzymes at concentrations 10X greater than achievable in solution. This membrane design maximizes the three steps governing the flux and selectivity of a CO₂ membrane: CO₂ capture (step I), HCO₃⁻ transport (step II), and CO₂ release (step III) enabling our enzymatic liquid membrane to exceed the Department of Energy standards for CO₂ sequestration technologies. The issue of novelty is now further addressed in the Discussion and Summary Section.

2. The enzymatic activity is affected by buffer pH and temperature, the relation of enzymatic activity and permeance could be investigated. The catalytic mechanism of CA for CO₂ should be also studied by enzyme kinetics.

Response: We would argue that determination of the catalytic mechanism of CA for CO₂ is outside the scope of the present paper as it has been studied in detail by other researchers (refs. 30-33). If the point is whether we tested the catalytic activity of the

immobilized CA in our membranes, we indirectly determined this by calculating the CO₂ flux based on the experimentally determined enzyme concentration per unit area (Page 24). The calculated flux was consistent with the average experimentally determined CO₂ flux, meaning that activity of the immobilized CA enzymes was comparable to or greater on average than the free CA in solution. This point is now made more clearly in Section “Enzymatic water-membrane performance” (Page 24).

3. Long-lasting stability is very important for the practical application of the membranes, so the long-term stability of the membranes should be tested.

Response: Agreed. To address the long-term stability, we now report our membrane performance data over a period of 3 months during storage at room temperature and 100% relative humidity conditions (see Supplementary Fig. S7). The only moderate loss of CO₂ permeance suggests that both water and enzymatic activity are largely retained.

3. In Figure 4, the orders of Figure a, b are not corresponding to the manuscript.

Response: Yes, thank you for pointing that out. We have completely revised the figures and captions to provide greater clarity.

4. How to control and obtain the actual amount of CA enzyme incorporated in each sample?

Response: The amount of CA is controlled by the depth and areal fraction of the 8 nm diameter hydrophilic pore channels that house the CA enzymes. In our case, the 8 nm diameter pores were designed to accommodate the ~6 nm diameter CA enzyme, and the 18-nm pore depth should allow only 2 or at most 3 CAs per pore channel. As we now show in Fig. 1 and the Supplementary Information, the Anodisc has ca. 70% porosity. Each Anodisc pore is filled with a close packed array of hydrophilic mesoporous silica pore channels (Fig. 1, and Supplementary Fig. S2). The hydrophilic channel area is calculated to be 3.92×10^{11} nanopores per cm² based on TEM measurements of the channel arrays (Fig. 1, Supplementary Fig. S2). If each channel houses two enzymes on average as determined by FTIR (Fig. 5e) that translates to 7.84×10^{11} CAs/cm² consistent with the CO₂ flux, if the CAs maintain their native enzymatic activity.

5. The authors should run a controlled experiment without CA enzymes. The membrane module and set-up for separation performance should be also provided for better understanding.

Response: Yes agreed, we performed this control experiment. As pointed out in the Materials and Methods Section Section, in control experiments, we measured the CO₂ and Ar permeance in the membrane prepared with hydrophilic channels containing water but no CA (prepared as in Fig. 2 Step 3). We observed no detectable CO₂ flux consistent with a defect free confined liquid-membrane. We also measured the CO₂ flux of a completely hydrophobic membrane (prepared as in Fig. 2 Step 2) without CA and measured a high flux without selectivity due to the open channels. In the SI, we now

show the two different set-ups for measurement of CO₂ flux and selectivity (Supplementary Fig. S3,5).

6. The introduction section is lengthy, and the authors should come to the point of enzyme based membranes.

Response: We tried to improve upon/shorten the introduction. However other reviewers are requesting that we cite some additional papers, so there is little net shortening.

We thank again the Editor and Reviewers for the careful review of our manuscript and for their comments and advice.

REVIEWERS' COMMENTS:

Reviewer #1 (Remarks to the Author):

I am satisfied with the revision of the manuscript carried out by the authors. I would hence like to recommend the manuscript for publication in Nature Communications.

Reviewer #2 (Remarks to the Author):

Bio-inspired ultra-thin, enzymatic, nano-stabilized liquid membrane for CO₂ capture and separation under ambient conditions

Yaqin Fu, Ying-Bing Jiang,* Darren Dunphy, Haifeng Xiong, Eric Coker, Stan Chou, Hongxia Zhang, Juan M. Vanegas, Jonas G. Croissant, Joseph L. Cecchi, Susan B Rempe, C. Jeffrey Brinker*

Nature Communications REVISION: NCOMMS-17-20912A

I think the authors have largely addressed many of the concerns raised by all the reviewers, including the comments I made as Reviewer #2. In particular, the Figures are greatly improved, and make the manuscript much more palatable. The authors did address the issues about practicality and scalability, which all the reviewers raised, although I still have a measure of scepticism about whether this really is practical or scalable. Overall, I think readers can learn something novel from the approach shown in the paper.

There are some minor points raised below, to address before I recommend publication of the article:

In Scheme 1c, the numbers 1, 2, and 3 appear for the transformations, but I didn't notice these mentioned or interpreted in the caption.

On the same page:

"containing 2 g l⁻¹ of carbonic anhydrase". I think the standard IUPAC abbreviation for liter is "L"

On the same page:

"However, the CO₂ permeability of the membrane (214×10⁻⁹) was exceptional"

Do the authors mean "permeance" or "permeability" here? If it is the latter, then the units of permeability need to be shown for the first time.

Reviewer #3 (Remarks to the Author):

This manuscript has been improved greatly after the revision and could be accepted for publication.

ANSWER TO THE REVIEWERS' COMMENTS

Reviewer #1 (Remarks to the Author)

I am satisfied with the revision of the manuscript carried out by the authors. I would hence like to recommend the manuscript for publication in Nature Communications.

Answer: We thank the reviewer for his time and expertise to examine our manuscript and enhance its quality.

Reviewer #2 (Remarks to the Author)

I think the authors have largely addressed many of the concerns raised by all the reviewers, including the comments I made as Reviewer #2. In particular, the Figures are greatly improved, and make the manuscript much more palatable. The authors did address the issues about practicality and scalability, which all the reviewers raised, although I still have a measure of skepticism about whether this really is practical or scalable. Overall, I think readers can learn something novel from the approach shown in the paper.

There are some minor points raised below, to address before I recommend publication of the article: In Scheme 1c, the numbers 1, 2, and 3 appear for the transformations, but I didn't notice these mentioned or interpreted in the caption.

Answer: We revised the figure and removed the numbers 1, 2, and 3 which were not necessary.

On the same page:

“containing 2 g l⁻¹ of carbonic anhydrase”. I think the standard IUPAC abbreviation for liter is “L”.

Answer: We agree with the reviewer that capital L could be used, but to follow the format guidelines of Nature Communications we use the alternative format.

On the same page:

“However, the CO₂ permeability of the membrane (214×10⁻⁹) was exceptional”

Do the authors mean “permeance” or “permeability” here? If it is the latter, then the units of permeability need to be shown for the first time.

Answer: Due to the need to substantially shorten the Introduction section, this sentence was

removed from the manuscript.

Reviewer #3 (Remarks to the Author)

This manuscript has been improved greatly after the revision and could be accepted for publication.

Answer: We thank the reviewer for his time and expertise to examine our manuscript and enhance its quality.

We thank again the Editor and Reviewers for the careful review of our manuscript and for their comments and advice.